# Antigingivitis and Antiplaque Effects of Oral Probiotic Containing the *Streptococcus salivarius* M18 Strain: A Randomized Clinical Trial

**DOI:** 10.3390/nu15183882

**Published:** 2023-09-06

**Authors:** Ksenia Babina, Dilara Salikhova, Vladlena Doroshina, Irina Makeeva, Alexandr Zaytsev, Matvey Uvarichev, Maria Polyakova, Nina Novozhilova

**Affiliations:** 1Department of Therapeutic Dentistry, I.M. Sechenov First Moscow State Medical University (Sechenov University), 119991 Moscow, Russia; salikhova_d_i@student.sechenov.ru (D.S.); doroshina_v_yu@staff.sechenov.ru (V.D.); makeeva_i_m@staff.sechenov.ru (I.M.); matvey.uvarichev@mail.ru (M.U.); polyakova_m_a_1@staff.sechenov.ru (M.P.); novozhilova_n_e@staff.sechenov.ru (N.N.); 2Institute of Linguistics and Intercultural Communication, I.M. Sechenov First Moscow State Medical University (Sechenov University), 119991 Moscow, Russia; zaytsev_a_b@staff.sechenov.ru

**Keywords:** dental plaque, gingivitis, probiotics, *Streptococcus salivarius* M18

## Abstract

We aimed to assess the effect of oral probiotic containing the *Streptococcus salivarius* M18 strain on gingival inflammation, bleeding on probing, and oral biofilm. Sixty-one consenting participants aged between 18 and 25 with gingivitis were recruited in this double-blind, parallel-group study and randomly divided into the probiotic group (*n* = 31) and the placebo group (*n* = 30). Fifty-seven participants completed the entire study protocol, 27 in the probiotic group and 30 in the placebo group. The outcomes were assessed after 4 weeks of intervention and 4 weeks of follow-up. There was a significant decrease in the Gingival Index, with the effect size of 0.58 [95%CI 0.05–1.10], and Turesky modification of the Quigley and Hein Plaque Index, with the effect size of 0.55 [95%CI: 0.02–1.07], in the probiotic group after the intervention. However, after a 4-week follow-up, the only significant treatment outcome was improved gingival condition according to the Gingival Index. The Gingival Bleeding Index also decreased significantly in the probiotic group after the intervention period; after the follow-up, this parameter did not differ significantly in both groups from the baseline values. In the placebo group, there were no significant improvements in the assessed parameters throughout this study. No serious side effects were registered. Within the limitations of this study, we conclude that the use of oral probiotic containing the *Streptococcus salivarius* M18 strain resulted in a significant improvement in gingival condition and oral hygiene level in young adults with gingivitis. Trial registration NCT05727436. Funding: none.

## 1. Introduction

Periodontal diseases are among the most common chronic diseases worldwide [1]. The prevalence of periodontal diseases is estimated at 78.8%, 90.7%, and 90.3% in adolescents, adults, and the elderly, respectively [2]. Periodontal inflammation can range from mild gingivitis to severe irreversible periodontitis, leading to tooth loss [3], and is considered a risk factor for certain systemic diseases [4]. Gingivitis is a reversible condition, which is believed to be a pre-existing stage of periodontitis. Thus, timely treatment of gingivitis is crucial to prevent periodontitis [5,6].

Current etiological concepts suggest that the dysbiotic growth of virulent microorganisms of dental plaque (biofilm) is a necessary component for the development of gingivitis and periodontitis [7,8,9,10]. The main bacteria causing gingivitis include some species of *streptococcus*, *fusobacterium*, *actinomyces*, *veillonella*, and *treponema*. *Bacteroides*, *capnocytophaga*, and *eikenella* are also likely to play a role in the etiology of gingivitis [11]. There are also individual factors such as smoking, endocrine disorders, medications, or systemic diseases that can alter the inflammatory response to plaque and thereby increase susceptibility to gingivitis [12].

Initial therapy for gingivitis consists of motivating and educating the patient, as well as mechanical removal of dental plaque [13,14]. However, most people do not properly control plaque accumulation with tooth brushing and interdental cleaning [5,15]. To overcome this, antimicrobial products have been suggested for their adjuvant efficacy to reduce plaque accumulation and gingivitis development [16]. At the same time, prolonged use of antiseptics can be associated with undesirable side effects, which has led to a search for alternative approaches [16]. Bacteriotherapy is a method in which a commensal strain is introduced into the host microbiome to rebalance the microflora [17]. This approach is of great interest for the treatment of oral diseases [18] due to minimal side effects [17]. It is potentially effective even if there is a lack of mechanical plaque control [19]. Its possible mechanisms of action are not fully understood. However, both direct and indirect effects seem to be evident [20]. Among them are coaggregation with pathogenic microorganisms and growth inhibition, production of bacteriocins and hydrogen peroxide, competition for adhesion sites of nutrient substrates, and systemic immunomodulation by affecting mucosal immune system cells or mucosal epithelial barrier function [21,22,23,24,25,26,27].

A number of studies have assessed the effects of probiotics in the prevention of dental caries [28,29,30,31,32] and the prevention and treatment of oral candidiasis [33,34,35], halitosis [36,37,38,39], and periodontal diseases [17,19,37,40,41,42]. Nase et al. were the first to suggest using probiotics in periodontal patients more than a decade ago [43]. However, the findings of existing studies remain contradictory. Some authors have reported a significant reduction in gingival and periodontal inflammation [40,42] and plaque accumulation, significant improvement in periodontal pocket depth [41], and reduction in pro-inflammatory cytokines [44]. At the same time, there have been reports of no positive effects on clinical and microbiological parameters [37,45] or of positive effects only on the composition of oral microbiota [46]. The heterogeneity of results can be explained by the variability in methodology and the use of different probiotic strains or their combinations [19]. The most popular microorganisms used as probiotics have been strains belonging to lacto- and bifidobacteria [47,48,49]. *Streptococcus salivarius* strains have been less studied but are also of great interest. The two main strains of *Streptococcus salivarius* used in dentistry are K12 and M18. *S. salivarius* K12 has been shown to provide an inhibitory effect on oral biofilm formation [50,51], and the use of *S. salivarius* M18 has been confirmed to reduce plaque index [52,53] and improve periodontal health indicators [53,54,55].

Still, healthcare professionals may be reluctant to recommend probiotics to patients due to a paucity of the peer-reviewed literature on the topic and the absence of accepted evidence-based recommendations [56,57]. A systematic review by Ausenda et al. concluded that more research on the topic was needed, as significant heterogeneity of the existing studies did not allow researchers to make firm conclusions and develop clinical recommendations regarding probiotics use in the treatment of periodontal diseases [58].

Therefore, the aim of our study was to expand the knowledge on the topic by assessing the effect of oral probiotic containing the *Streptococcus salivarius* M18 strain on gingival inflammation, bleeding on probing, and oral biofilm.

## 2. Materials and Methods

### 2.1. Study Design

This study’s protocol was approved by the local ethics committee (Protocol no. 23-22, 17 November 2022) and was registered on clinicaltrials.gov (NCT05727436, February 2023). This study complied with the 1964 Declaration of Helsinki and its subsequent amendments and the CONSORT 2010 statement.

This was a double-blind, randomized, placebo-controlled, two-arm parallel-group trial evaluating the effect of oral probiotic containing *Streptococcus salivarius* M18 strain on the gingival inflammation, bleeding on probing, and oral biofilm. This study was conducted between February 2023 and May 2023 at the Therapeutic Dentistry department of Sechenov University, Moscow, Russia.

### 2.2. Subjects

This study included healthy young adults visiting Sechenov University’s Dental Institute. The enrollment was accomplished by the single trained operator (D.S.). All participants were informed of the purpose and implications of this study and signed written informed consent. At baseline, all patients were instructed to follow a standardized brushing technique (Bass) using a pea-sized amount of toothpaste without antibacterial or anti-plaque components twice a day.

The inclusion criteria were the following:-men and women aged between 18 and 25;-permanent bite;-presence of at least 20 teeth;-absence of systemic or chronic diseases;-the diagnosis of gingivitis stated clinically.

The exclusion criteria were the following:-refusal to provide written informed consent;-chronic periodontitis;-use of medication or dietary supplement that contain pro- and prebiotics (within 1 month before the enrollment);-use of antibiotics (within 3 months before the enrollment);-allergy to the components of the probiotic/placebo;-immunodeficiency disorders and use of immunosuppressants;-use of other hygiene products, immunostimulants, antibacterial drugs, and pre-and probiotics during this study;-failure to follow the prescription regimen;-failure to follow the research protocol.

### 2.3. Randomization, Blinding, and Interventions

Sixty-one participants were randomly assigned to two groups in accordance with a computer-generated sequence prepared by a third-party person. Group 1 received lozenges containing probiotic; Group 2 received lozenges containing placebo. Allocation concealment was performed by a person not involved in this study using numbered unlabeled containers. Subjects and researchers were unaware of which participants received probiotics and who received placebos. The oral probiotic tested in this study was *S. salivarius* M18 ≥ 5 × 10^8^ CFU per lozenge, mint-flavored gray-and-white lozenges (Dentoblis, MEDICO DOMUS, d.d.o., Nis, Serbia). The placebo lozenges did not contain *S. salivarius* M18 but had the same flavor, consistency, and appearance.

Participants were instructed to take one lozenge daily in the evening after toothbrushing, let it dissolve, not bite or swallow it, and then avoid eating or drinking for at least one hour. To increase participants’ compliance, they were asked to mark each lozenge intake on Google spreadsheets.

### 2.4. Endpoints and Assessments

A one-month intervention period was followed by a one-month follow-up.

Primary endpoints included changes in the Gingival Index (GI) and Gingival Bleeding Index (GBI) after the one-month intervention period and after the one-month follow-up. Secondary endpoints were changes in the Turesky modification of the Quigley and Hein Plaque Index (TQHPI) values after the one-month intervention period and after the one-month follow-up. The parameters were assessed as described in previous studies [59,60,61].

### 2.5. Statistical Analyses

#### 2.5.1. Sample Size Calculation

The sample size was determined for the primary outcome (GBI) according to our pilot study. Sample size calculations were performed using the G*Power calculator (version 3.1.9.6) for the Wilcoxon rank sum test for the two groups—the power was set at 80%, and the alfa-level was set as 0.05. The allocation ratio was equal to 1. The resultant target sample size comprised 30 participants in each group (26 participants according to sample size calculations plus 15% to account for possible dropout), 60 patients in total.

#### 2.5.2. Statistical Methods

Continuous data were presented as means and standard deviations, 95% confidence intervals, and medians and quartiles; categorical variables were presented as counts and percentages. The normality and sphericity of distribution were assessed with a Shapiro–Wilk test and Levene’s test, respectively. Fisher’s exact test was used to compare gender distribution between this study’s groups. A repeated analysis of variance and a paired *t*-test or Welch *t*-test were used to compare GI and TQHPI values between this study’s groups. A Wilcoxon rank sum test and a Wilcoxon matched-pairs signed-rank test were used to compare the GBI values between this study’s groups. Hedge’s g was used to calculate the effect size between the groups by comparing mean baseline and post-intervention values. Collected data were analyzed in R version 3.6.0 (26 April 2019) with the following packages: “doBy”, “rstatix”, “stats”, and “effectsize” in RStudio version 1.2.1335 2009–2019. Statistical significance was defined as *p*  <  0.05 (two-tailed).

## 3. Results

Two hundred and five patients aged between 18 and 25 were assessed for eligibility. Of these, 144 patients were ineligible due to the following reasons: not meeting inclusion criteria (*n* = 84); meeting exclusion criteria (*n* = 55); refusing to participate (*n* = 1); and not being sure they would comply with the protocol (*n* = 4). Sixty-one patients were enrolled and randomly assigned to the probiotic group (21 females and 10 males) or the placebo group (19 females and 11 males). Four participants from the probiotic group were lost to the follow-up due to a self-reported allergy (*n* = 1), measles quarantine (*n* = 2), and failure to check up (*n* = 1). A total of 57 participants were included in the final analysis, 27 in the probiotic group and 30 in the placebo group (Figure 1). Group comparisons showed no significant differences in age and gender distribution (Table 1).

We assessed the effect of the probiotic intake on gingival inflammation (GI and GBI) and plaque accumulation (TQHPI).

Participants receiving the probiotic showed a significant decrease in gingival bleeding at the post-intervention time point. The mean GBI values were 0.127 ± 0.137 and 0.086 ± 0.086 at baseline and at 4 weeks. Participants receiving the placebo, on the contrary, showed an increase in this parameter from 0.066 ± 0.049 to 0.083 ± 0.052 (Table 2). However, the GBI values after the follow-up period in both groups did not differ significantly from the baseline values.

According to ANOVA, the “visit” (*p* = 0.012) and interaction of “visit” and “group” factors (*p* < 0.001) had a significant impact on the GI values (Table 3).

The probiotic group showed lower mean post-intervention and post-follow-up GI values than the placebo group (*p* = 0.045). According to the GI, a 4-week probiotic intake resulted in an improvement of gingival condition with the effect size of 0.58; moreover, post-follow-up GI values did not differ significantly from post-intervention values. The placebo group showed a significant increase in post-intervention GI values, while post-follow-up values did not differ significantly from baseline and post-intervention values (Table 4 and Figure 2).

The mean trajectory showed a decrease in TQHPI values over the 4-week intervention period in the probiotic group (Figure 3), and interaction between the factors indicated significant differences between the groups (Table 5 and Table 6). According to this parameter, a 4-week probiotic intake resulted in a reduction in plaque scores with the effect size of 0.55. No significant changes in TQHPI values were registered in the placebo group throughout this study.

### Adverse Events

No serious adverse events were reported. One patient from the probiotic group canceled the participation in this study due to a self-reported allergy (skin rash). However, the patient did not attend the visit scheduled to assess his condition; thus, the allergic reaction was not confirmed.

## 4. Discussion

We assessed the effect of oral probiotic containing the *Streptococcus salivarius* M18 strain on gingival inflammation (GI), bleeding on probing (GBI), and oral biofilm (TQHPI). A 4-week probiotic intake resulted in a significant improvement in gingival condition and oral hygiene level. However, after a 4-week follow-up, the only significant treatment outcome was an improved gingival condition, according to the GI.

Gingivitis is defined as “an inflammatory lesion resulting from interactions between the dental plaque biofilm and the host’s immune–inflammatory response, which remains contained within the gingiva and does not extend to the periodontal attachment” [62]. It is characterized by redness, swelling, and bleeding of the gingiva [63]. Therefore, the assessment of gingival status should include a subjective assessment of the color, form, density, and bleeding tendency of the gingival tissues, as well as oral hygiene performance expressed as qualitative or semi-quantitative indices [64]. In our study, we used the GI to assess gingival changes in color and texture and the GBI to assess bleeding elicited by probing.

Most previous studies assessing the effect of probiotics on gingival health indicators used *lactobaccilli* or *bifidobacteria* strains in different formulations. In a study by Toiviainen involving adults, the probiotic lozenges containing a combination of *Lactobacillus rhamnosus* GG and *Bifidobacterium animalis* subsp. *lactis* BB-12 decreased GI values, while no changes were observed in the control group after 4 weeks [65]. In a similar study involving adolescents, a reduction in GI was registered in both groups; however, it was significantly higher in the probiotic group than in the control group [66]. According to Twetman et al., the number of sites with bleeding on probing decreased in the test and placebo groups, but the changes were significant only in the test groups (after one or two weeks of chewing *Lactobacillus reuteri*-containing gum) [67]. Keller et al. assessed bleeding on probing after a 2- and 4-week probiotic intake (*Lactobacillus rhamnosus* and *Lactobacillus curvatus*) and a 2-week follow-up period. They found that the decrease in this parameter was statistically significant after 4 and 6 weeks in the test group compared to baseline [17]. In a study by Schlagenhauf et al., the observed mean GI score in the test group that took *Lactobacillus reuteri*-containing lozenges for 6 weeks significantly decreased, whereas in the placebo group, it significantly increased compared to baseline values. As the participants were recruited among sailors of a naval ship on a mission at sea, the decrease in the GI could be attributed to a deterioration of oral hygiene [19].

Some of the previous studies confirmed the beneficial effects of *lactobaccilli* or *bifidobacteria* strains on periodontal parameters in patients with different health problems. Yuki et al. found that the consumption of milk fermented with *Lactobacillus rhamnosus* by individuals with intellectual disabilities resulted in a decrease in GI scores during this study, with a tendency for a greater decrease in the test group. Also, there was a significantly greater reduction in PMA index scores in the test group compared to the placebo group [68]. Sabatini et al. assessed the effect of oral probiotics (*Lactobacillus reuteri*) in the management of gingivitis in diabetic patients. In their study, at 30 days, both groups showed a statistically significant reduction in bleeding on probing; the changes were more pronounced in the probiotic group [69]. In a study by Hambire and Hambire, there was a significant improvement in the gingival status of children undergoing chemotherapy in the test group after the 90-day use of oral probiotics [70].

In other studies, the effects of probiotic use on gingival health were questionable. Benic et al. showed that GI values were almost identical in the test (*Streptococcus salivarius* M18) group and the placebo group at baseline, at the end of the intervention (1 month), and at a 3-month follow-up [37]. Montero et al. evaluated the potency of oral tablets containing *Lactobacillus plantarum*, *Lactobacillus brevis*, and *Pediococcus acidilactici* in the treatment of gingivitis. Both study groups experienced a statistically significant improvement in mean GI scores. At 6 weeks, despite insignificant differences between the groups in general, GI values for the sites of severe inflammation (GI = 3 at baseline) were significantly lower in the probiotic group [63].

In our study, we used an oral probiotic containing the *Streptococcus salivarius* M18. To the best of our knowledge, the literature on the use of Streptococci strains for dental purposes is scarce. A recent systematic review on the use of probiotics in the treatment of periodontal diseases, which included 21 studies, showed that the majority of the studies used lactobacilli and bifidobacteria, whereas Streptococci species were used only in one study [71]. Another similar systematic review, which included 64 studies, had similar results, with Streptococci used only in three studies [47].

Burton et al. reported no improvement and no significant differences in GI scores between the probiotic-treated (*Streptococcus salivarius* M18) children and the controls at 1-, 3-, and 7-month time points [52]. In a study by Babina et al., the authors assessed the effect of probiotic containing the *Streptococcus salivarius* K12 strain on oral health indicators. Although PMA values tended to decrease in the probiotic group at the end of a 4-week treatment period and after a 2-week washout period, these changes did not reach the level of statistical significance [50]. Similar results were obtained by Ferrer et al., who assessed the effect of a bucco-adhesive gel containing *Streptococcus dentisani* applied in a dental splint for 5 min every 48 h for a period of 1 month. Oral health indicators were assessed at baseline, 15 and 30 days after the first application, and 15 days after the end of treatment. The GI decreased in both groups throughout this study; however, the changes were more pronounced in the probiotic group than in the placebo at each of the visits, but the differences were insignificant [23].

In our study, we found that a 4-week intake of probiotic resulted in an improvement of the GI with the effect size of 0.58; moreover, this effect remained after a 4-week follow-up period. The probiotic group showed a significant decrease in GI and GBI scores at the post-intervention time point; the placebo group, on the contrary, showed an increase in these parameters. However, after the follow-up period, the GBI values in both groups did not differ significantly from the baseline.

The antigingivitis effect of probiotics demonstrated in a number of studies could be explained by a decrease in oral biofilm accumulation. In clinical settings, the extent of oral biofilm accumulation may be assessed using oral hygiene indices. Many studies reported a reduction in plaque scores according to hygienic indices alongside a reduction in gingival inflammation [18,19,70]. In our study, we observed a significant reduction in TQHPI scores in the probiotic group with the effect size of 0.55, while no significant changes were registered in the placebo group.

Several studies have also evaluated the anti-plaque and direct anti-inflammatory effects of oral probiotics in patients with experimental gingivitis [18,72,73] and reported controversial results. Hallström et al. assessed the effect of probiotic containing *Lactobacillus reuteri* on gingival health in experimental gingivitis. They concluded that daily intake of probiotic lozenges did not seem to significantly affect plaque accumulation, inflammatory reaction, or composition of the biofilm during experimental gingivitis [72].

Kuru et al. compared gingival indices in the placebo and probiotic (*bifidobacterium*-containing yogurt) groups at baseline, after 28 days of this study product usage, and subsequently, after plaque accumulation. After plaque accumulation (a 5-day non-brushing period), significantly better results for plaque and gingivitis scores and bleeding on probing were registered in the probiotic group compared to the control group [18]. Lee et al. reported that *Lactobacillus brevis* CD2 may delay gingivitis development in the model of experimental gingivitis by downregulating the inflammatory cascade. Their study suggests that the antigingivitis effect of probiotics may be associated not only with a decrease in plaque accumulation but also with a direct anti-inflammatory effect [73].

All in all, the studies on the use of probiotics in periodontal patients were different in methodology and, therefore, are very difficult to compare. The reported effects of probiotics could be strain-specific, product-specific, disease-specific, group-specific, etc. However, in general, current evidence is supportive of the use of probiotics in managing gingivitis or periodontitis [58].

We readily acknowledge several limitations to our study. We focused on clinical parameters of gingival health and biofilm accumulation without the assessment of microbiological ones, taking into consideration that clinically relevant outcomes are more important. Also, we enrolled patients of a limited age group (18–25-year-olds), as within this age limit, the majority of patients do not exhibit attachment loss (i.e., have gingivitis, not periodontitis). Further studies with longer intervention periods are needed, as a four-week intake of probiotics is a relatively short period for the colonization of the oral cavity with a tested microorganism [63].

## 5. Conclusions

According to our findings, the intake of oral probiotic with the *Streptococcus salivarius* M18 strain resulted in a significant improvement in the gingival condition and oral hygiene levels in young adults with gingivitis. However, taking into consideration the limitations of our study, its results should be generalized with caution, and more studies with broader age groups and longer intervention periods are needed to assess the long-term effect of *Streptococcus salivarius* M18-containing probiotics.

## Figures and Tables

**Figure 1 nutrients-15-03882-f001:**
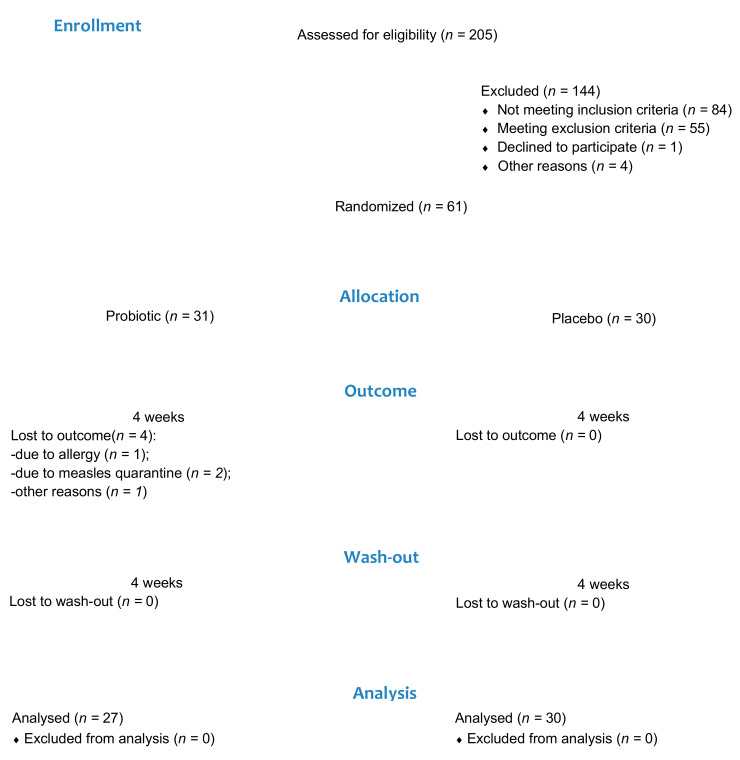
Patient flow diagram.

**Figure 2 nutrients-15-03882-f002:**
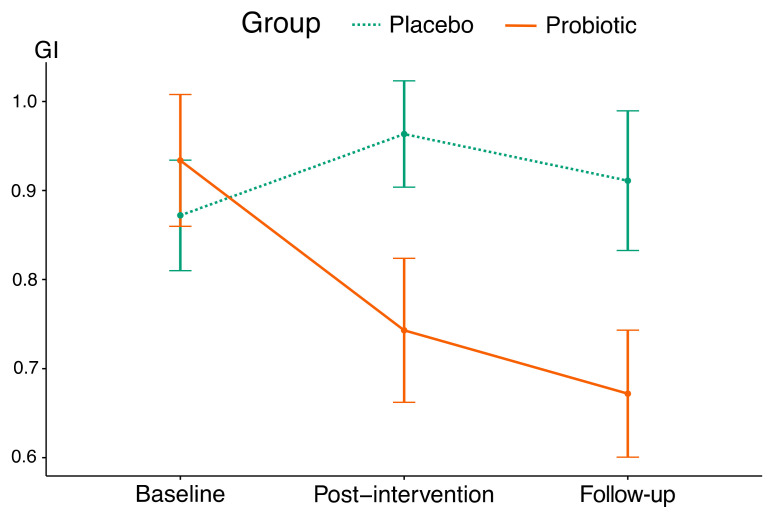
The mean trajectories of GI values in this study’s groups (vertical lines indicate standard deviations).

**Figure 3 nutrients-15-03882-f003:**
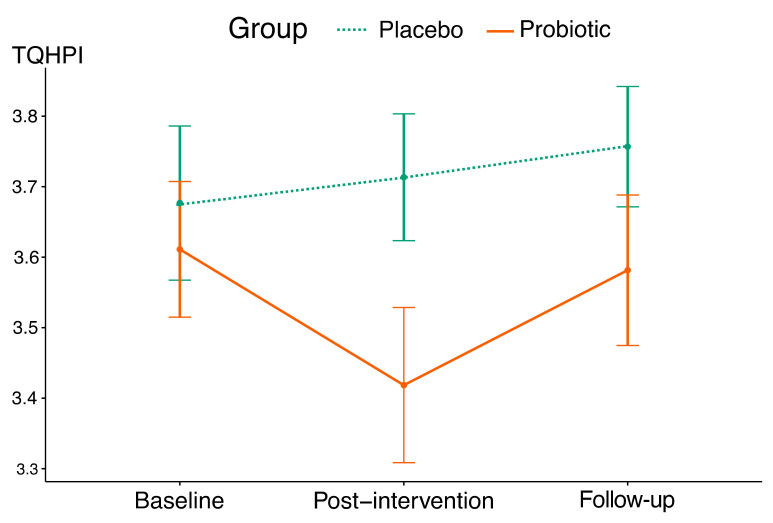
The mean trajectories of TQHPI values in this study’s groups (vertical lines indicate standard deviations).

**Table 1 nutrients-15-03882-t001:** Subject demographics.

Group	Probiotic (*n* = 27)	Placebo (*n* = 30)	*p*-Value
Sex, *n* (%)			
Female	18 (66.7)	19 (63.3)	1.0 ^1^
Male	9 (33.3)	11 (36.7)	
Total	27 (100)	30 (100)	
Age			
Mean (sd)	21.7 (1.9)	21.1 (1.4)	0.2563 ^2^
Median (Q1, Q3)	21 (20, 23)	21 (20, 21.75)	
Min, Max	19, 26	20, 25	

^1^ Fisher’s exact test. ^2^ Wilcoxon rank sum test.

**Table 2 nutrients-15-03882-t002:** GBI values.

Group	Probiotic (*n* = 27)	Placebo (*n* = 30)	Between-Group Comparisons	Effect Size (Hedges’ g) ^2^
Baseline				
Mean (SD)	0.127 (0.137)	0.066 (0.049)		
95%CI	0.08–0.18	0.05–0.08	*p* = 0.1217 ^1^	
Median (Q1, Q3)	0.09 (0.045, 0.18)	0.05 (0.025, 0.11)		−0.04
	^a^	^A^		95%CI: −0.55–0.48
Post-intervention				
Mean (SD)	0.086 (0.086)	0.083 (0.052)		
95%CI	0.05–0.12	0.06–0.1	*p* = 0.5057 ^1^	
Median (Q1, Q3)	0.07 (0.03, 0.12)	0.07 (0.0425, 0.12)		
	^b^	^B^		
Follow-up				
Mean (SD)	0.108 (0.093)	0.094 (0.047)	*p* = 0.9553 ^1^	
95%CI	0.07–0.14	0.08–0.11		
Median (Q1, Q3)	0.09 (0.05, 0.145)	0.085 (0.06, 0.1275)		
	^a^	^AB^		
∆ (Post-intervention–Baseline)				
Mean (SD)	−0.04 (0.1)	0.02 (0.03)	*p* = 0.0003648 ^1^	
95%CI	−0.08–0.00	0.00–0.03		
Median (Q1, Q3)	−0.02 (−0.06, 0.00)	0.01 (0.00, 0.03)		
∆ (Follow-up–Baseline)				
Mean (SD)	0.02 (0.03)	0.01 (0.03)	*p* = 0.2192 ^3^	
95%CI	0.01–0.03	0.00–0.02		
Median (Q1, Q3)	0.02 (0.00, 0.05)	0.01 (−0.01, 0.03)		

^a, b, A, B^ Different letters indicate statistically significant differences between the time points according to Wilcoxon matched-pairs signed-rank tests. ^1^ Wilcoxon rank sum test. ^2^ Effect size was calculated between baseline and post-intervention values. ^3^ Welch *t*-test.

**Table 3 nutrients-15-03882-t003:** ANOVA table for GI indices values.

Factor	DFn	DFd	F-Value	*p*-Value
Group	1.00	55.00	2.076	0.155
Visit	1.72	94.82	4.973	0.012 *
Group * Visit	1.72	94.82	11.333	0.0000931 *

* statistically significant values. DFn–degree of freedom for the numerator of the F ratio. DFd–degree of freedom for the denominator of the F ratio.

**Table 4 nutrients-15-03882-t004:** GI values.

	Probiotic (*n* = 27)	Placebo (*n* = 30)	Statistical Analysis	Effect Size (Hedges’ g) ^2^
Baseline				
Mean (SD)	0.934 (0.385)	0.872 (0.339)	*p* = 0.523 ^1^	
95%CI	0.79–1.08	0.75–0.99		0.58
Median (Q1, Q3)	0.96 (0.71, 1.06)	0.875 (0.71, 1.07)		95%CI: 0.05–1.10
	^a^	^A^		
Post-intervention				
Mean (SD)	0.743 (0.419)	0.963 (0.327)	*p* = 0.045 ^1^	
95%CI	0.58–0.9	0.85–1.08		
Median (Q1, Q3)	0.67 (0.42, 1.00)	1.02 (0.71, 1.17)		
	^b^	^B^		
Follow-up				
Mean (SD)	0.672 (0.370)	0.911 (0.429)	*p* = 0.045 ^1^	
95%CI	0.53–0.81	0.76–1.06		
Median (Q1, Q3)	0.67 (0.375, 0.92)	0.94 (0.64, 1.17)		
	^b^	^AB^		
∆ (Post-intervention–Baseline)				
Mean (SD)	−0.19 (0.23)	0.09 (0.2)	*p* < 0.001 ^3^	
95%CI	−0.28–−0.11	0.02–0.16		
Median (Q1, Q3)	−0.13 (−0.33, −0.08)	0.09 (−0.05, 0.13)		
∆ (Follow-up–Baseline)				
Mean (SD)	−0.07 (0.34)	−0.05 (0.27)	*p* = 0.5921 ^3^	
95%CI	−0.2–0.06	−0.15–0.05		
Median (Q1, Q3)	−0.04 (−0.15, 0.11)	0.00 (−0.09, 0.09)		

^a, b, A, B^ Different letters indicate statistically significant differences between the time points. ^1^ ANOVA. ^2^ Effect size was calculated between baseline and post-intervention values. ^3^ Wilcoxon rank sum test.

**Table 5 nutrients-15-03882-t005:** ANOVA table for TQHPI indices values.

Factor	DFn	DFd	F-Value	*p*-Value
Group	1.00	55	1.805	0.1805
Visit	2	110	3.363	0.038 *
Group * Visit	2	110	0.025	0.025 *

* significant values. DFn—degree of freedom for the numerator of the F ratio. DFd—degree of freedom for the denominator of the F ratio.

**Table 6 nutrients-15-03882-t006:** TQHPI index values.

	Probiotic (*n* = 27)	Placebo (*n* = 30)	Statistical Analysis	Effect Size (Hedges’ g) ^2^
Baseline				
Mean (SD)	3.60 (0.50)	3.67 (0.59)	*p* = 0.657 ^1^	
95%CI	3.42–3.8	3.46–3.89		
Median (Q1, Q3)	3.66 (3.283, 3.9555)	3.586 (3.2815, 4.13125)		0.55
	^a^	^A^		95%CI: 0.02–1.07
Post-intervention				
Mean (SD)	3.42 (0.57)	3.71 (0.49)	*p* = 0.123 ^1^	
CI95%	3.2–3.63	3.54–3.89		
Median (Q1, Q3)	3.5 (3.0625, 3.8025)	3.702 (3.30175, 4.0045)		
	^b^	^A^		
Follow-up				
Mean (SD)	3.58 (0.56)	3.75 (0.47)	*p* = 0.302 ^1^	
CI95%	3.37–3.79	3.59–3.92		
Median (Q1, Q3)	3.714 (3.3215, 4.009)	3.613 (3.473, 4.08475)		
	^ab^	^A^		
∆ (Post-intervention–Baseline)				
Mean (SD)	−0.19 (0.38)	0.04 (0.23)	*p* = 0.005569 ^1^	
95%CI	−0.33–−0.05	−0.05–0.12		
Median (Q1, Q3)	−0.20 (−0.35, 0.05)	0.00 (−0.10, 0.10)		
∆ (Follow-up–Baseline)				
Mean (SD)	0.16 (0.39)	0.04 (0.19)	*p* = 0.1517 ^3^	
95%CI	0.02–0.31	−0.02–0.11		
Median (Q1, Q3)	0.10 (−0.10, 0.25)	0.00 (−0.1, 0.2)		

^a, b, A^ Different letters indicate statistically significant differences between the time points. ^1^ ANOVA. ^2^ Effect size was calculated between baseline and post-intervention values. ^3^ Welch *t*-test.

## Data Availability

The datasets used and/or analyzed during the current study are available from the corresponding author upon reasonable request.

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
