# Peer review of "Antigingivitis and Antiplaque Effects of Oral Probiotic Containing the Streptococcus salivarius M18 Strain: A Randomized Clinical Trial"

_nutrients, 2023, doi:10.3390/nu15183882_

Round 1
Reviewer 1 Report
Overall, nicely described RCT clearly described, but some revisions needed.
1. Firstly, all throughout the text and in all references, note that the SPECIES is NOT capitalized and both genus and species should be in italics.
2. Introduction, line 48. Please define ‘bad habits’, this could be anything. Give examples.
3. Introduction, line 68-69. Need to reference missing citations (#20,22,27) that assessed probiotics for dental caries.
4. Minor formatting under 2.2. Subjects. Need to add tabs line 122 and 127 so 2nd sentence is under the preceding sentence.
5. Section 2.3, line 132 please define dose 5 x 10(8) per lozenge?
6. Figure 1. Missing boxes around text and arrows. Please fix.
7. Replace “wash-out” with follow-up period. Usually wash-out period refers to cross-over trials.
8. Add to Tables 1,2,4,6. You need to provide an additional row and provide the MEAN CHANGE from baseline to end of follow-up for each group. Readers should not have to do these calculations themselves.
9. Tables 1,2,4,6. Use of a, A, a, B a AB is confusing in the tables and don’t really relate to which p values?? Clarify or leave out (unnecessary).
10. Figure 2 and 3. In Black and White the different lines can not be differentiated. Use a square line indicator or dashed line for one. Too hard to read (too small).
11. Adverse events. Nicely described.
12. Discussion. Delete 2nd paragraph (unneeded), line 242-243.
13. Discussion. Line 297. Reference #78 is very old (2016 meta-analysis). There are >20 meta-analyses on probiotics and dental infections published 2022-2023. Use more recent ones. Example: Hardan L 2022 Pharmceu or Gheisarn Z 2022 Nutrients).
14. Discussion. Missing a statement on how(if) your study results may/may not be generalized to another population.
Author Response
Dear Reviewer,
We want to thank you most sincerely for your time and effort spent to review our article and for your valuable comments. We believe that corrections made in accordance with your comments will improve the article and make it clearer and more useful for the readers.
The article was revised in accordance with your recommendations.
- Firstly, all throughout the text and in all references, note that the SPECIES is NOT capitalized and both genus and species should be in italics.
We have corrected it throughout the manuscript.
- Introduction, line 48. Please define ‘bad habits’, this could be anything. Give examples.
We have changed “bad habits” to “smoking”.
- Introduction, line 68-69. Need to reference missing citations (#20,22,27) that assessed probiotics for dental caries.
We have added these links to the studies assessing periodontal health.
- Minor formatting under 2.2. Subjects. Need to add tabs line 122 and 127 so 2nd sentence is under the preceding sentence.
We have corrected this.
- Section 2.3, line 132 please define dose 5 x 10(8) per lozenge?
We have added this information.
- Figure 1. Missing boxes around text and arrows. Please fix.
We have corrected it.
- Replace “wash-out” with follow-up period. Usually wash-out period refers to cross-over trials.
We have corrected it.
- Add to Tables 1,2,4,6. You need to provide an additional row and provide the MEAN CHANGE from baseline to end of follow-up for each group. Readers should not have to do these calculations themselves.
We have added the mean changes to the tables.
- Tables 1,2,4,6. Use of a, A, a, B a AB is confusing in the tables and don’t really relate to which p values?? Clarify or leave out (unnecessary).
Thank you for your comment. These letters indicate statistically significant differences within each group between the study timepoints. We have corrected the typo in the footnotes.
- Figure 2 and 3. In Black and White the different lines can not be differentiated. Use a square line indicator or dashed line for one. Too hard to read (too small).
We have changed the figures.
- Adverse events. Nicely described.
Thank you.
- Discussion. Delete 2nd paragraph (unneeded), line 242-243.
We have deleted it.
- Discussion. Line 297. Reference #78 is very old (2016 meta-analysis). There are >20 meta-analyses on probiotics and dental infections published 2022-2023. Use more recent ones. Example: Hardan L 2022 Pharmceu or Gheisarn Z 2022 Nutrients).
We have corrected this.
- Discussion. Missing a statement on how(if) your study results may/may not be generalized to another population.
We have added this statement to the conclusion of the study.
Sincerely yours,
Authors
Reviewer 2 Report
The manuscript entitled "Antigingivitis and Antiplaque Effects of Oral Probiotics Containing the Streptococcus Salivarius M18 Strain: A Randomized Clinical Trial" is interesting, relevant, and presents in a well-structured manner the effect of oral probiotics containing the Streptococcus salivarius M18 17 strain. The trial is registered with the number NCT05727436.
The authors presented the aims of their research.
The manuscript presents scientifically sound results with reproducible results based on the details given in the methods section.
Materials and methods are clearly presented, including the treatment duration (weeks) and the treatment characteristics (of oral probiotics containing the Streptococcus salivarius M18-17 strain).
The gingival inflammation, bleeding on probing, and oral biofilm of the studied patients were identified after the clinical examination.
The results presented the correlation between applied therapy and outcomes.
The presented tables are appropriate.
The discussion section also provides the results of other authors on the studied problem, but most of the citations are more than 5 years old.
The conclusions are clearly presented and argued.
The cited references are relevant, but of a total of 80, many references are older than 5 years (the bibliographic titles 3, 4, 6–17, 19, 23, 24, 25–32, 36–40, 45, 47–55, 58, 65, 67–69, 71, 73, 75, 78, 80).
For the reasons given above, I consider that the manuscript deserves to be published, but after major corrections relating to the cited references.
Author Response
Dear Reviewer,
Thank you very much for your relevant and important recommendations. We deeply appreciate your help and interest in our work.
The cited references are relevant, but of a total of 80, many references are older than 5 years (the bibliographic titles 3, 4, 6–17, 19, 23, 24, 25–32, 36–40, 45, 47–55, 58, 65, 67–69, 71, 73, 75,
We have added latest references in accordance with your recommendations. Now 80% of the references are not older than 5 years. However, we would like to keep some of the references from the original version of the manuscript to ensure coherence and not to leave out the meaningful studies related to our field of interest.
Sincerely yours,
Authors
Round 2
Reviewer 2 Report
Dear authors,
I have checked the final form of the manuscript, and I congratulate you on the work done.
The authors presented the aims of their research.
The manuscript presents scientifically sound results with reproducible results based on the details given in the methods section.
Materials and methods are clearly presented, including the treatment duration (weeks) and the treatment characteristics (of oral probiotics containing the Streptococcus salivarius M18–17 strain).
The gingival inflammation, bleeding on probing, and oral biofilm of the studied patients were identified after the clinical examination.
The results presented the correlation between applied therapy and outcomes.
The presented tables are appropriate.
Now, the discussion section also provides the results of other authors on the studied problem, and most of the citations are current.
The conclusions are clearly presented and argued.
Now, the cited references are relevant.
For the reasons given above, I consider that the manuscript deserves to be published in this form.
We wish you all the best!
Author Response
Dear Reviewer,
We want to thank you most sincerely for your time and effort spent to review our article and for your appreciation of our work.
Sincerely yours,
Authors